# Supplementary Light with Increased Blue Fraction Accelerates Emergence and Improves Development of the Inflorescence in *Aechmea*, *Guzmania* and *Vriesea*

Elahe Javadi Asayesh [1], Sasan Aliniaeifard [1,*], Naser Askari [2], Mahmood Reza Roozban [1], Mohammadhadi Sobhani [1], Georgios Tsaniklidis [3], Ernst J. Woltering [4,5] and Dimitrios Fanourakis [6]

[1] Photosynthesis Laboratory, Department of Horticulture, Aburaihan Campus, University of Tehran, Tehran 33916-53755, Iran; Elahejavadi@ut.ac.ir (E.J.A.); mroozban@ut.ac.ir (M.R.R.); m.h.sobhani@ut.ac.ir (M.S.)

[2] Department of Horticultural Science, University of Jiroft, Jiroft 78671-55311, Iran; naser.askari@ujiroft.ac.ir

[3] Hellenic Agricultural Organization ELGO-DIMITRA, Institute of Olive Tree, Subtropical Crops and Viticulture, 73134 Chania, Greece; tsaniklidis@elgo.iosv.gr

[4] Wageningen Food & Biobased Research, Bornse Weilanden 9, 6708 WG Wageningen, The Netherlands; Ernst.Woltering@wur.nl

[5] Horticulture & Product Physiology, Wageningen University, Droevendaalsesteeg 1, 6708 PB Wageningen, The Netherlands

[6] Laboratory of Quality and Safety of Agricultural Products, Landscape and Environment, Department of Agriculture, School of Agricultural Sciences, Hellenic Mediterranean University, Estavromenos, 71004 Heraklion, Greece; dfanourakis@hmu.gr

\* Correspondence: aliniaeifard@ut.ac.ir

**Abstract:** In protected cultivation, increasing the light level via supplementary lighting (SL) is critical to improve external quality, especially in periods with low light availability. Despite wide applications, the effect of light quality remains understated. In this study, the effect of SL quality and nutrient solution electrical conductivity (EC) on growth and flowering of three bromeliad species was investigated. Treatments included solar light, and this supplemented with R90B10 [90% red (R) and 10% blue (B)], R80B20 (80% R and 20% B), and R70B30 (70% R and 30% B). These were combined with an EC of 1 and 2 dS m⁻¹. Irrespective of the light treatment, the higher EC promoted growth, inflorescence emergence, and development in *Aechmea fasciata* (Lindl.) Baker, whereas adverse effects were noted in *Guzmania* and *Vriesea*. The higher EC-induced negative effect in *Guzmania* and *Vriesea* was slightly alleviated by SL. With few notable exceptions, SL exerted limited effects on photosynthetic functionality. Depending on the species, SL improved external quality traits. In all species, SL increased root and inflorescence weight and stimulated biomass allocation to generative organs. It also accelerated inflorescence emergence and promoted inflorescence development. In this way, the time to commercial development stage was considerably shortened. These effects were more prominent at R80B20 and R70B30. Under those conditions, for instance, inflorescence emergence occurred 3–5 weeks earlier than in the control, depending on the species. In conclusion, SL with increased B proportion leads to shorter production period owing to faster emergence and improved development of the inflorescence and is recommended for commercial use.

**Keywords:** biomass partitioning; bromeliad; chlorophyll fluorescence imaging; electrical conductivity; LED; O–J–I–P-transient; production period

## 1. Introduction

Bromeliads (Bromeliaceae) are widely cultivated as ornamental pot plants due to exuberant foliage, peculiar flower characteristics, the potential for year-round flowering, and sustaining bloom for long periods (weeks to months) under interior conditions [1–3]. The most commercialized bromeliad species belong to the genera *Aechmea*, *Guzmania*, *Neoregelia*, *Tillandsia*, and *Vriesea*. Among them, *Guzmania* and *Vriesea* cultivars account for

60 and 15% of annual pot plant sales, respectively [4,5]. In Central and South America, *Aechmea* species maintain high popularity [2].

In commercial horticulture, uniform and scheduled flowering of bromeliads is conventionally achieved by using ethylene releasing agents, such as acetylene [6,7]. Still, long cultivation is required for the production of flowering bromeliad plants [8]. For instance, the phase between flower induction and anthesis lasts 68 and 114 d for *Aechmea fasciata* and *Vriesea* 'splendens,' respectively, with the total cultivation period being 14 and 12 months [9]. Supplementary light is often employed not only to shorten the production period but also to improve the customer-perceived (visual) quality of several ornamental species [10,11]. For instance, a higher light intensity decreased the flower induction period (by 10–70 d) depending on the bromeliad species [12,13]. Moreover, higher light intensity was associated with increased (25%) biomass [14], and improved inflorescence quality (i.e., size and number of florets; [13]). Although the beneficial effect of increased light intensity on the above-mentioned aspects is well-documented, limited attention has been devoted to the role of light quality (spectral distribution). With the development of light-emitting diodes (LEDs), it is now readily feasible to manipulate light quality [15,16]. The interest in using this possibility to promote plant growth and external quality traits is currently expanding [17,18].

Photosynthetic performance is affected by light quality during cultivation [15,16]. Chlorophyll fluorescence is commonly employed for the non-invasive assessment of the electron transport system efficiency [19]. Polyphasic chlorophyll fluorescence induction curves are a novel means of extracting additional information on the photosynthetic apparatus structure and function [17].

Although bromeliads typically require low nutrient solution electrical conductivity (EC; $\approx 1$ dS m$^{-1}$), a higher EC (1.5 dS m$^{-1}$) stimulated plant growth in some species. Additionally, plants cultivated under supplemental light often require an enriched nutrient supply [20]. Therefore, changes in light quality may necessitate an adjusted nutrient intake depending on the species of interest.

In order to fully take advantage of the light quality manipulation, as well as to provide guidance for commercial applications in ornamental horticulture, the role of EC alongside light quality was investigated. The objectives of this study were to assess the light regime and EC effects on plant growth, morphology, and biomass partitioning, as well as on flower emergence and development. In addition, chlorophyll fluorescence parameters were obtained by using two protocols to evaluate the impact of the growth regime on photosynthetic performance. To determine whether or not the treatment effects were limited to a specific $CO_2$ assimilation pathway, species (*Aechmea fasciata* 'Primera', *Guzmania* 'Rostara' and *Vriesea* 'Splenriet') using different types of photosynthesis were employed. The selected species are all equipped with a phytotelm (the so-called tank), which retains water by intercepting precipitation, and in this way, facilitates plant hydration at periods of low soil water availability [21,22].

## 2. Materials and Methods

### 2.1. Plant Material and Growth Conditions

In the present study, *Aechmea fasciata* 'Primera' (crassulacean acid metabolism; [23,24]), *Guzmania* 'Rostara' (three carbon photosynthesis; [25]), and *Vriesea* 'Splenriet' (three carbon photosynthesis; [24]) were employed. Un-rooted transplants were obtained from a commercial nursery (Corn. Bak B.V., Assendelft, the Netherlands). The ones with uniform height (~ 15 cm) and architecture were selected for potting. For each species, the pot size for commercial purposes was used. This was 8 cm for *Guzmania* and *Vriesea*, and 12 cm for *Aechmea*. Pots were filled with peat moss and perlite mixture (1:3, *v/v*). Pots were then transferred in a multi-span plastic greenhouse, which was located to Pakdasht (Tehran Province, Iran; 35°28'51″ N 51°41'05″ E). Following 4 months of cultivation, plants were fully rooted and treatments were initiated. Treatments lasted between January and September of 2019. Air temperature and relative air humidity were recorded with sensors (Sensohive Technologies

ApS, Odense, Denmark) installed in the upper third of the plant canopy. During cultivation, air temperature ranged between 17.5 and 26.5 °C, while relative air humidity between 50 and 76%. The average solar daily light integral was $6.4 \pm 0.5$ mol m$^{-2}$ d$^{-1}$.

Two factors (4 light regimes × 2 EC levels) were applied in combination. Plants were grown under solar light (non-lighted control), or supplemented with 3 combinations of red (R) and blue (B) light by using LED modules (Guangzhou Grow Light Company Model IGL-158-18R17B7-DC; Input voltage: 220–240 V; 18 W; 0.09 A). These were R90B10 [90% R (peak at 660 nm) and 10% B (peak at 450 nm)], R80B20 (80% R and 20% B), and R70B30 (70% R and 30% B). Supplemental light was provided daily at a photosynthetic photon flux density of 120 µmol m$^{-2}$ s$^{-1}$ at the canopy level for 12 h (0800 to 2000 h). In this way, the photoperiod was extended during the intervals of January to April (by 22–165 min), and August to September (by 5 and 47 min, respectively). Light intensity was determined using a PAR-FluorPen device (FP 100-MAX, Photon Systems Instruments, Drasov, Czech Republic), and the spectrum by a SpectroMaster (SEKONIC C-7000, Tokyo, Japan). To avoid light contamination, opaque black-white plastic films were placed around each light regime treatment. To minimize border effects, plants adjacent to these films were not sampled.

Nutrient solution EC was set to 1 or 2 dS m$^{-1}$ (further referred as EC1 and EC2, respectively). The nutrient solution composition for EC1 was based on a commercial source (Corn. Bak B.V., Assendelft, the Netherlands) (Table 1). For EC2, the concentration of the macro-elements was doubled, maintaining the same ratio as in the original recipe (Table 1). Nutrient solution pH was adjusted to 5.75 [26,27].

**Table 1.** The composition of nutrient solutions with electrical conductivity (EC) of 1 and 2 dS m$^{-1}$ (EC1 and EC2, respectively) was employed in the current study.

| Element | Concentration (mmol L$^{-1}$) | |
|:---:|:---:|:---:|
| | **EC1** | **EC2** |
| K | 6.78 | 13.56 |
| Mg | 0.83 | 1.66 |
| $NO_3^-$ | 7.48 | 14.96 |
| $NH_4$ | 1.25 | 2.5 |
| $PO_4^{3-}$ | 0.55 | 1.1 |
| $SO_4^{2-}$ | 0.837 | 1.674 |
| Fe | 0.013 | |
| Mn | 0.0026 | |
| Zn | 0.0024 | |
| Cu | 0.0025 | |
| Na | 0.00004 | |
| Mo | 0.00004 | |

Plants were irrigated twice a week [26]. To prevent element accumulation, the substrate was washed with distilled water once per month.

Sampling was conducted at the end of the growth period. Sampled leaves were young, fully-expanded, and grown under direct light. Replicate leaves were collected from separate plants. In all instances, the time between sampling and the start of the evaluation was less than 15 min.

*2.2. Flower Induction*

Six months following the onset of the experiment, flower induction was performed based on commercial practices [26]. Briefly, saturated acetylene solution (18–20 °C) was placed in the tank of the plants. This solution (0.05 dS m$^{-1}$ EC and pH of 6.9) was prepared by injecting acetylene gas (0.5 bar for 12 min) into 20 L of water. The acetylene treatment was applied in the morning (08:00–10:00 h; 17–19 °C) and repeated a week later.

Two weeks before the first treatment and 4 weeks following the second treatment, nutrient solution supply was ceased [9,12]. To secure flower induction, plants were kept fully irrigated 14 d before the first treatment. Additionally, 3 d before each treatment, irrigation was performed in a way that the tanks were full of water, and the substrate was completely wet. Plants were not irrigated for 2 d after each treatment.

### 2.3. Plant Growth, Morphology, and Biomass Allocation

Nine months following the onset of the experiment, plant growth, morphology, and biomass allocation were determined. Evaluations included plant height (from the root-to-shoot junction to the apical inflorescence end), number of offshoots (the so-called pups), crown thickness (1 cm above the root-to-shoot junction), tank volume, number of leaves, and leaf area. For leaf area assessment, leaves were scanned (HP Scanjet G4010, Irvine, CA, USA) and then evaluated by using the Digimizer software (version 4.1.1.0, MedCalc Software, Ostend, Belgium).

Following removal of the substrate from the roots via gentle washing, root volume was measured by employing a volume-displacement technique [28]. Plant roots were suspended in a cylinder filled with water. Root volume was then determined by measuring the volume of water displaced by the plant roots.

Leaf, root, and inflorescence (fresh and dry) masses were also recorded. For measuring dry weight, samples were placed in a forced-air drying oven for 72 h at 80 °C. By using dry mass, specific leaf area (SLA; leaf area/leaf mass), flower mass ratio (FMR; flower mass/plant mass), leaf mass ratio (LMR; leaf mass/plant mass), and root mass ratio (RMR; root mass/plant mass) were calculated. All the measurements were conducted on 4 plants per treatment.

### 2.4. Inflorescence Emergence and Development

In bromeliads, the first sign of inflorescence induction in the tank was considered as the beginning of flowering. Inflorescence development was recorded from induction (induced by the first acetylene treatment) until harvest (≈3 month following induction, and 9 months following the onset of the experiment). Inflorescence development was scored based on the scale (1 to 5) depicted in Figure 1. Evaluations were conducted on 4 plants per treatment.

### 2.5. Leaf Osmotic Potential

To maintain cell turgor pressure, plant response to increased nutrient supply often includes osmotic potential adjustments [29]. On this basis, leaf sap osmotic potential was evaluated. For sap extraction, leaves were divided into small segments and placed along with 2 metal spheres (1 mm diameter) in tubes perforated with 4 holes (0.5 mm diameter). Each tube was then encased in a larger one, and centrifuged ($15,000\times g$) for 15 min. For evaluation, sap extract was collected from the larger tube. Osmolarity was assessed with a vapor pressure osmometer (Osmomat 030, Gonotec GmbH, Berlin, Germany). The unit conversion was performed by using the Van't Hoff equation [osmotic potential (MPa) = − osmolarity (mosmoles $kg^{-1}$) × 2.58 × $10^{-3}$].

### 2.6. SPAD Value and Leaf Photosynthetic Pigment Content

Leaf pigment content is affected by the growth environment and has implications for both photosynthetic capacity and pot plant ornamental value [30,31]. In this perspective, the leaf SPAD value and photosynthetic pigment (chlorophyll, carotenoids) content were assessed.

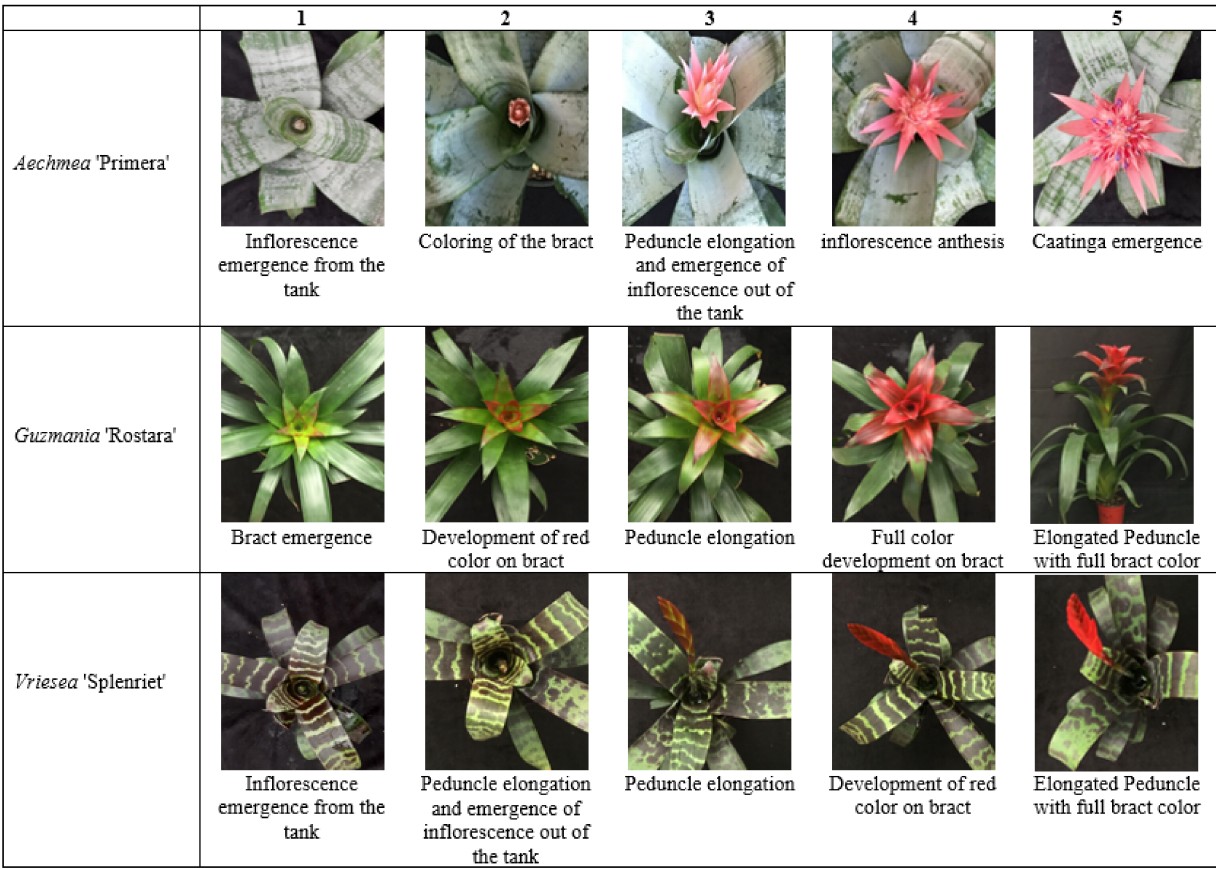

**Figure 1.** Scale characterizing inflorescence development in the three species under study.

Leaves were measured by using a SPAD-502 (Konica Minolta Corp., Solna, Sweden). This instrument provides a non-invasive approximation of leaf chlorophyll content by considering leaf transmittance in red and near-infrared light spectra (650 and 940 nm, respectively; [32]). Three points were recorded per replicate leaf and were further averaged. Three replicate leaves were assessed per treatment.

The same leaves were employed for photosynthetic pigment content. Following fine chopping, portions weighing 0.5 g were homogenized with the addition of 10 mL of 80% acetone. This primary acetone extract was then filtered, and the filtered extract was diluted by adding 2 mL of 80% acetone per mL of extract. Since chlorophyll is light sensitive, extraction took place in a dark room [33]. The obtained extract was subjected to reading on a spectrophotometer (Mapada UV-1800; Shanghai. Mapada Instruments Co., Ltd., Shanghai, China). Total chlorophyll and carotenoid contents were calculated [34]. Four leaves were assessed per treatment.

## 2.7. Chlorophyll Fluorescence Imaging

As a sensitive indicator of plant photosynthetic performance, dark-adapted values of the maximum quantum yield of PSII ($F_v/F_m$) were recorded in detached leaves of each treatment. Measurements were conducted by using a FluorCam FC 1000-H (Photon Systems Instruments, Brno, Czech Republic). Leaves were dark-adapted ($\geq$20 min) prior to evaluation. Then, $F_v/F_m$ was evaluated by applying a saturated photosynthetic photon flux density of 3900 µmol m$^{-2}$ s$^{-1}$ [11,19]. Four leaves were assessed per treatment.

## 2.8. Polyphasic Chlorophyll Fluorescence Transient (OJIP) Evaluation

A polyphasic chlorophyll fluorescence induction curve (O–J–I–P-transient) was obtained in detached leaves of each treatment. By employing the JIP test, the shape changes of the OJIP transient were quantitatively translated to a set of parameters, which relate



to the in vivo adaptive behavior of the photosynthetic apparatus (especially PSII) to the growth environment [17]. Measurements were conducted by using a PAR-fluorPen FP 100-MAX (Photon Systems Instruments) following dark adaptation ($\geq$20 min). These were obtained at intervals of 50 μs, as well as of 3, 30, and 300 ms. The employed light intensity (3900 mmol m$^{-2}$ s$^{-1}$ photosynthetic photon flux density) was sufficient to generate maximal fluorescence for all treatments. Based on the OJIP protocol-obtained data, the performance index for the photochemical activity (PI$_{ABS}$) was calculated [35]. Light curves were also obtained by exposing the leaves to different photosynthetic photon flux densities [0 (darkness), 100, 200, 300, 500, and 1000 μmol m$^{-2}$ s$^{-1}$]. Four leaves were assessed per treatment.

### 2.9. Experimental Design and Statistical Analysis

A completely randomized split plot design with 2 factors (light regime, EC level) was realized. Since the pot size and growth pattern of the employed species were different, each was analyzed separately. Data analyses were carried out using the SAS statistical software (SAS Institute, Cary, NC, USA). Data were tested for homogeneity of variances (Duncan's test). Subsequently, estimated least significant differences (LSD) of treatment effects were determined ($p$ = 0.05).

## 3. Results

### 3.1. Plant Growth, Morphology, and Biomass Allocation

Supplementary light increased plant height in *Aechmea* and *Guzmania* (Tables 2 and 3). In the former species the highest plant height was under R80B20 (Table 2), while in the latter plant height became higher as the percentage of B increased (Table 3). Supplementary light generally increased the number of offshoots in *Aechmea* (Table 2). Supplementary light generally increased crown thickness in *Guzmania* (Table 3). Supplementary light enhanced tank volume in *Guzmania*, with R70B30 exerting a considerable effect (Table 3). Higher EC decreased tank volume in *Aechmea* and *Guzmania* (Tables 2 and 3). In the absence of supplementary light, higher EC also induced yellow spots on the leaves of *Aechmea* and *Guzmania* (data not shown).

Supplementary light decreased SLA (a proxy of leaf thickness) in *Guzmania* (Table 3), while higher EC decreased SLA in *Aechmea* (Table 2). Supplementary light increased leaf area in *Vriesea*, whereas higher EC decreased it (Table 4). Supplementary light increased leaf dry weight in *Guzmania* and *Vriesea* (Tables 3 and 4). Higher EC increased leaf dry weight in *Aechmea* (Table 2) and decreased it in *Vriesea* (Table 4).

Supplementary light increased root dry weight in all three species (Tables 2–4). Higher EC decreased root dry weight in *Guzmania* (Table 3) and *Vriesea* (Table 4).

Supplementary light drastically increased flower dry weight in all three species (Tables 2–4). Higher EC increased flower dry weight in *Aechmea* (Table 2) and decreased it in *Guzmania* (Table 3) and *Vriesea* (Table 4).

Supplementary light considerably increased the fresh mass partitioning to the inflorescences in all three species (Figure 2). R80B20 and R70B30 were generally associated with increased fresh mass partitioning to the inflorescences as compared to R90B10 (Figure 2).

### 3.2. Inflorescence Emergence and Development

Supplementary light decreased the time required for inflorescence emergence (Figure 3). R80B20 and R70B30 were associated with a shorter time for inflorescence emergence as compared to R90B10 (Figure 3). Higher EC increased the time required for inflorescence emergence of *Guzmania* (Figure S1).

**Table 2.** Effect of supplementary light and nutrient solution electrical conductivity (EC) on growth and morphology of *Aechmea* 'Primera' plants. The light treatment included control (no supplementary light), R90B10 [90% red (R) and 10% blue (B)], R80B20 (80% R and 20% B), and R70B30 (70% R and 30% B), while the EC values employed were 1 and 2 dS m$^{-1}$. Four replicate plants were assessed per treatment. In traits, where the interaction of the two factors (light regime, EC) was significant, different letters indicate significant differences. FW, fresh weight; DW, dry weight.

| EC (dS m$^{-1}$) | Light Regime | Plant Height (cm) | Number of Offshoots | Crown Thickness (mm) | Tank Volume (mL) | SLA (cm$^2$ g$^{-1}$) | Leaf | | | | Root | | | Flower | |
|---|---|---|---|---|---|---|---|---|---|---|---|---|---|---|---|
| | | | | | | | Number | Area (cm$^2$) | FW (g) | DW (g) | FW (g) | DW (g) | Volume (cm$^3$) | FW (g) | DW (g) |
| 1 | Control | 42.1 | 0.50 | 32.8 | 25.9 | 112.5 | 16.0 | 3680.5 | 344.2 c | 35.7 | 40.3 | 4.59 | 47.0 | 27.6 | 1.77 |
| | R90B10 | 43.3 | 0.75 | 32.7 | 18.8 | 108.6 | 14.5 | 3838.5 | 366.7 bc | 40.2 | 63.3 | 7.90 | 62.8 | 80.6 | 7.11 |
| | R80B20 | 48.3 | 3.50 | 35.9 | 38.8 | 125.9 | 12.3 | 3508.5 | 348.8 bc | 32.2 | 60.5 | 8.36 | 67.0 | 110.1 | 10.68 |
| | R70B30 | 46.5 | 1.50 | 32.3 | 29.0 | 110.9 | 14.3 | 3953.2 | 351.6 bc | 40.8 | 56.8 | 6.90 | 57.5 | 110.6 | 11.92 |
| 2 | Control | 39.8 | 0.50 | 32.5 | 25.3 | 106.2 | 16.8 | 3123.5 | 334.6 c | 33.5 | 35.3 | 4.39 | 39.0 | 44.5 | 3.29 |
| | R90B10 | 45.0 | 2.50 | 32.1 | 19.3 | 100.2 | 16.0 | 4244.3 | 419.2 a | 49.1 | 49.5 | 6.46 | 52.0 | 70.8 | 7.35 |
| | R80B20 | 50.8 | 1.75 | 34.5 | 22.0 | 90.50 | 14.5 | 3816.5 | 412.4 a | 48.2 | 54.3 | 6.41 | 59.3 | 127.6 | 15.23 |
| | R70B30 | 45.5 | 2.75 | 35.5 | 23.5 | 107.8 | 16.3 | 4103.7 | 395.1 ab | 44.7 | 57.9 | 6.23 | 70.0 | 110.2 | 12.82 |
| *p* value | Light regime | 0.014 * | 0.028 * | 0.247 ns | 0.113 ns | 0.905 ns | 0.2 ns | 0.165 ns | 0.109 ns | 0.124 ns | 0.039 * | 0.003 ** | 0.047 * | 0.0002 ** | 0.0001 ** |
| | EC | 0.88 ns | 0.527 ns | 0.877 ns | 0.038 * | 0.034 * | 0.046 * | 0.703 ns | 0.0004 ** | 0.013 * | 0.252 ns | 0.053 ns | 0.501 ns | 0.321 ns | 0.035 * |
| | Light regime × EC | 0.61 ns | 0.097 ns | 0.636 ns | 0.093 ns | 0.203 ns | 0.89 ns | 0.399 ns | 0.024 * | 0.749 ns | 0.768 ns | 0.423 ns | 0.375 ns | 0.257 ns | 0.246 ns |

ns = non-significant. Significance at the 0.05 probability level is indicated by *, and significance at the 0.01 probability level by **.

**Table 3.** Effect of supplementary light and nutrient solution electrical conductivity (EC) on growth and morphology of *Guzmania* 'Rostara' plants. The light treatment included control (no supplementary light), R90B10 [90% red (R) and 10% blue (B)], R80B20 (80% R and 20% B), and R70B30 (70% R and 30% B), while the EC values employed were 1 and 2 dS m⁻¹. Four replicate plants were assessed per treatment. In traits, where the interaction of the two factors (light regime, EC) was significant, different letters indicate significant differences. FW, fresh weight; DW, dry weight.

| EC (dS m⁻¹) | Light Regime | Plant Height (cm) | Number of Offshoots | Crown Thickness (mm) | Tank Volume (mL) | SLA (cm² g⁻¹) | Leaf | | | | Root | | | Flower | |
|---|---|---|---|---|---|---|---|---|---|---|---|---|---|---|---|
| | | | | | | | Number | Area (cm²) | FW (g) | DW (g) | FW (g) | DW (g) | Volume (cm³) | FW (g) | DW (g) |
| 1 | Control | 31.9 | 0.00 | 14.1 | 3.55 | 180.7 | 23.5 b | 1490.7 c | 58.8 c | 8.85 | 12.4 | 2.49 | 13.3 | 25.7 | 2.97 |
| | R90B10 | 37.5 | 0.25 | 18.4 | 5.45 | 155.0 | 26.0 ab | 1939.9 ac | 79.7 ab | 13.97 | 19.7 | 4.27 | 22.0 | 49.0 | 6.15 |
| | R80B20 | 38.3 | 0.00 | 17.8 | 5.10 | 167.7 | 27.3 a | 2217.4 ab | 80.6 ab | 14.07 | 22.9 | 5.05 | 26.5 | 46.2 | 6.00 |
| | R70B30 | 41.3 | 0.25 | 18.5 | 15.75 | 152.7 | 28.0 a | 2310.8 a | 98.5 a | 16.47 | 22.1 | 4.50 | 23.5 | 59.7 | 8.02 |
| 2 | Control | 28.8 | 0.00 | 13.5 | 3.05 | 168.0 | 26.5 ab | 1498.4 c | 66.1 bc | 9.53 | 10.4 | 1.91 | 11.0 | 5.88 | 0.62 |
| | R90B10 | 36.9 | 0.50 | 17.5 | 3.40 | 147.6 | 28.8 a | 2043.9 ab | 99.2 a | 14.77 | 17.9 | 3.17 | 16.8 | 19.1 | 2.05 |
| | R80B20 | 37.6 | 0.50 | 16.1 | 4.87 | 150.8 | 27.0 ab | 1875.0 abc | 83.8 ab | 13.77 | 19.3 | 3.77 | 19.8 | 44.5 | 5.26 |
| | R70B30 | 42.6 | 0.25 | 19.3 | 9.37 | 138.3 | 26.3 ab | 1762.1 bc | 82.0 ab | 13.44 | 21.5 | 3.59 | 20.3 | 47.6 | 5.48 |
| *p* value | Light regime | 0.002 ** | 0.573 ns | 0.02 * | 0.0001 ** | 0.008 ** | 0.377 ns | 0.048 * | 0.028 * | 0.006 ** | 0.002 ** | 0.004 ** | 0.004 ** | 0.0001 ** | 0.0001 ** |
| | EC | 0.58 ns | 0.299 ns | 0.666 ns | 0.012 * | 0.095 ns | 0.136 ns | 0.032 * | 0.394 ns | 0.503 ns | 0.119 ns | 0.0007 ** | 0.003 ** | 0.005 ** | 0.001 ** |
| | Light regime × EC | 0.698 ns | 0.703 ns | 0.939 ns | 0.054 ns | 0.97 ns | 0.037 * | 0.046 * | 0.039 * | 0.208 ns | 0.846 ns | 0.701 ns | 0.573 ns | 0.266 ns | 0.303 ns |

ns = non-significant. Significance at the 0.05 probability level is indicated by *, and significance at the 0.01 probability level by **.

**Table 4.** Effect of supplementary light and nutrient solution electrical conductivity (EC) on growth and morphology of *Vriesea* 'Splenriet' plants. The light treatment included control (no supplementary light), R90B10 [90% red (R) and 10% blue (B)], R80B20 (80% R and 20% B), and R70B30 (70% R and 30% B), while the EC values employed were 1 and 2 dS m$^{-1}$. Four replicate plants were assessed per treatment. In traits, where the interaction of the two factors (light regime, EC) was significant, different letters indicate significant differences. FW, fresh weight; DW, dry weight.

| EC (dS m$^{-1}$) | Light Regime | Plant Height (cm) | Crown Thickness (mm) | Tank Volume (mL) | SLA (cm$^2$ g$^{-1}$) | Leaf | | | | Root | | | Flower | |
|---|---|---|---|---|---|---|---|---|---|---|---|---|---|---|
| | | | | | | Number | Area (cm$^2$) | FW (g) | DW (g) | FW (g) | DW (g) | Volume (cm$^3$) | FW (g) | DW (g) |
| 1 | Control | 16.8 | 18.3 a | 9.5 | 135.0 | 14.3 | 748.9 | 51.0 | 5.86 | 3.25 | 1.31 | 4.5 | 7.13 | 0.69 |
| | R90B10 | 16.5 | 19.0 a | 13.0 | 148.5 | 15.8 | 989.0 | 55.7 | 7.30 | 3.13 | 1.67 | 4.5 | 13.8 | 1.61 |
| | R80B20 | 18.5 | 18.9 a | 21.3 | 130.5 | 16.3 | 1158 | 69.3 | 9.53 | 3.27 | 2.23 | 6.1 | 23.6 | 2.57 |
| | R70B30 | 17.0 | 18.3 a | 23.8 | 133.6 | 16.0 | 935.4 | 56.9 | 7.52 | 3.25 | 1.75 | 5.8 | 22.6 | 2.15 |
| 2 | Control | 14.8 | 14.3 b | 4.25 | 132.2 | 10.5 | 488.3 | 31.3 | 3.88 | 1.27 | 0.52 | 2.6 | 0 | 0 |
| | R90B10 | 15.5 | 20.5 a | 21.3 | 124.7 | 13.0 | 675.9 | 45.0 | 5.73 | 3.25 | 0.79 | 3.1 | 4.97 | 0.39 |
| | R80B20 | 16.8 | 17.7 a | 7.75 | 138.7 | 14.0 | 744.8 | 47.1 | 5.91 | 3.76 | 1.36 | 4.0 | 11.7 | 1.01 |
| | R70B30 | 17.3 | 18.5 a | 19.5 | 142.9 | 14.3 | 935.3 | 55.0 | 7.09 | 3.52 | 1.16 | 3.5 | 14.3 | 1.20 |
| *p* value | Light regime | 0.408 ns | 0.009 ** | 0.093 ns | 0.969 ns | 0.045 * | 0.004 ** | 0.017 * | 0.003 ** | 0.083 ns | 0.033 * | 0.182 ns | 0.005 ** | 0.003 ** |
| | EC | 0.105 ns | 0.181 ns | 0.403 ns | 0.694 ns | 0.0002 ** | 0.001 ** | 0.003 ** | 0.003 ** | 0.412 ns | 0.0005 ** | 0.0005 ** | 0.001 ** | 0.0003 ** |
| | Light regime × EC | 0.615 ns | 0.042 * | 0.381 ns | 0.192 ns | 0.562 ns | 0.133 ns | 0.252 ns | 0.217 ns | 0.064 ns | 0.92 ns | 0.878 ns | 0.867 ns | 0.529 ns |

ns = non-significant. Significance at the 0.05 probability level is indicated by *, and significance at the 0.01 probability level by **.

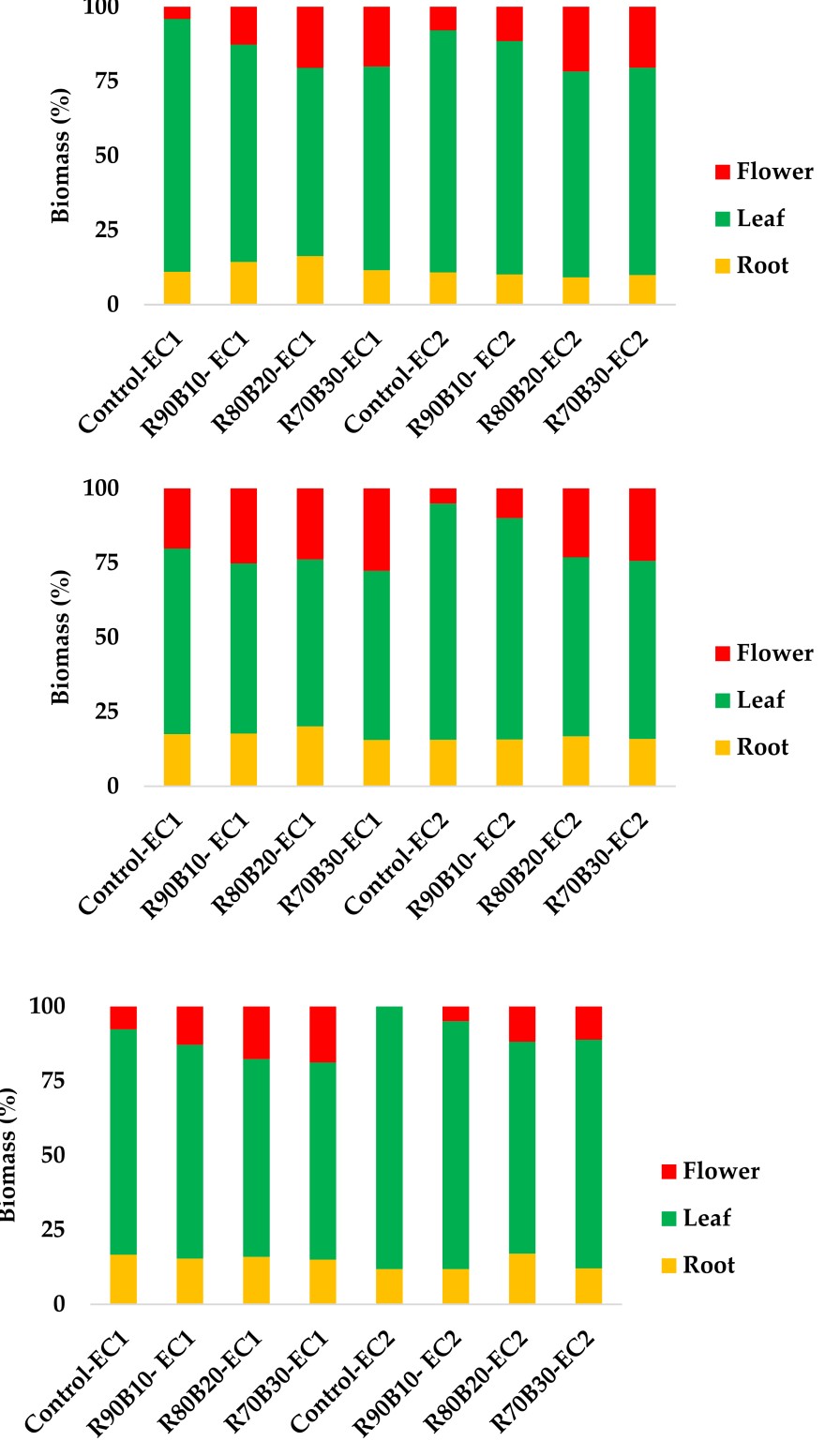

**Figure 2.** Effect of supplementary light and nutrient solution electrical conductivity (EC) on biomass partitioning of *Aechmea* 'Primera' (**top** panel), *Guzmania* 'Rostara' (**middle** panel), and *Vriesea* 'Splenriet' (**bottom** panel) plants. The light treatment included control (no supplementary light), R90B10 [90% red (R) and 10% blue (B)], R80B20 (80% R and 20% B), and R70B30 (70% R and 30% B), while the EC values employed were 1 and 2 dS m$^{-1}$. Four replicate plants were assessed per treatment. Statistics are provided in Table S1.

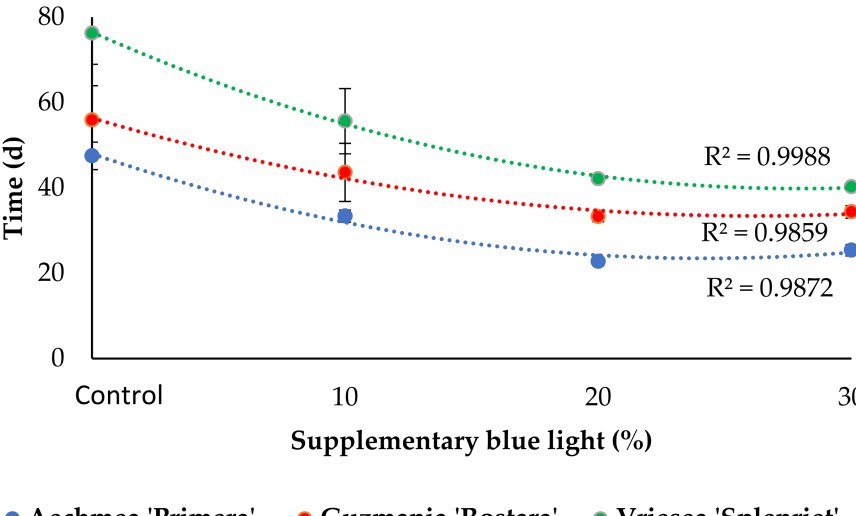

● **Aechmea 'Primera'**   ● **Guzmania 'Rostara'**   ● **Vriesea 'Splenriet'**

**Figure 3.** Time to inflorescence emergence following acetylene treatment as a function of the supplementary light percentage of blue in the three species under study. Supplementary light included R90B10 [90% red (R) and 10% blue (B)], R80B20 (80% R and 20% B), and R70B30 (70% R and 30% B). Control plants did not receive supplementary light. Data of the two EC treatments were pooled. Eight replicate plants were assessed per treatment. Error bars represent SEM.

Supplementary light increased inflorescence development (Figure 4). In *Aechmea* and *Guzmania*, R80B20 and R70B30 were associated with enhanced inflorescence development as compared to R90B10 (Figure 4). Higher EC promoted inflorescence development in *Aechmea*, whereas it decreased it in *Guzmania* and *Vriesea* (Figure S2).

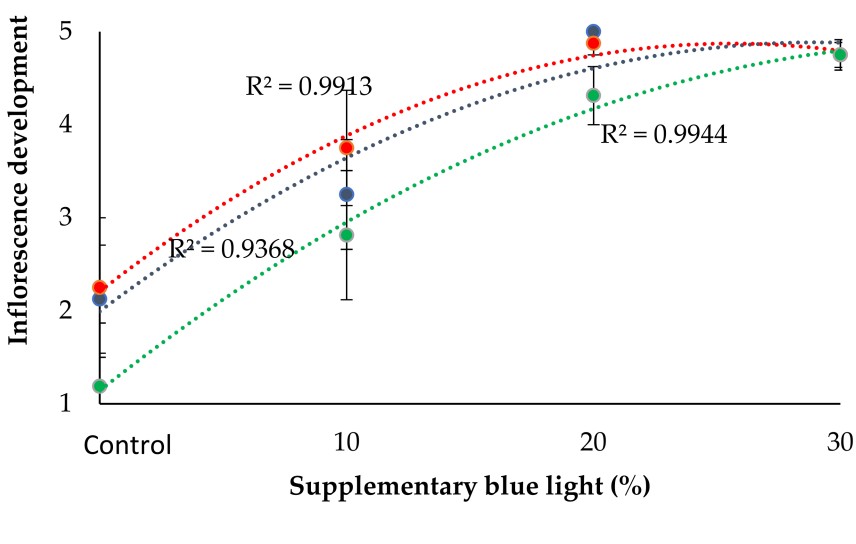

● **Aechmea 'Primera'**   ● **Guzmania 'Rostara'**   ● **Vriesea 'Splenriet'**

**Figure 4.** Inflorescence development as a function of the supplementary light percentage of blue in the three species under study. Supplementary light included R90B10 [90% red (R) and 10% blue (B)], R80B20 (80% R and 20% B), and R70B30 (70% R and 30% B). Control plants did not receive supplementary light. Data of the two EC treatments were pooled. The scale (1 to 5) characterizing inflorescence development is provided in Figure 1. Eight replicate plants were assessed per treatment. Error bars represent SEM.

### 3.3. Leaf Osmotic Potential

In *Aechmea*, leaf osmotic potential of plants fed by EC1 and exposed to R80B20 and R70B30 was higher (i.e., less negative) as compared to those fed by EC2 and exposed to supplemental light (Table S2). Higher EC decreased leaf osmotic potential (i.e., more negative) in *Guzmania* and *Vriesea* (25.9 and 68.4%, respectively; Tables S3 and S4).

### 3.4. SPAD Value and Leaf Photosynthetic Pigment Content

In *Aechmea*, the SPAD value was affected by the light regime, with plants grown under R70B30 having the highest value (Table S2). In *Guzmania*, the SPAD value was affected by both the light and EC regimes, with the highest values noted under R70B30 and EC2 (Table S3). In *Vriesea*, an interaction between light and EC regimes was apparent, where plants grown under R70B30 had higher SPAD value as compared to other light regimes at EC2 (Table S4).

In *Aechmea*, leaf chlorophyll and carotenoid contents were affected by both the light and EC regimes (Table S2). Cultivation under R70B30 promoted chlorophyll content, whereas it decreased carotenoid content. Instead, the higher EC decreased (8%) chlorophyll content, and stimulated (37.5%) carotenoid content. In *Guzmania* and *Vriesea*, light and EC regimes affected neither chlorophyll nor carotenoid contents (Tables S3 and S4).

### 3.5. Chlorophyll Fluorescence Imaging

In *Aechmea*, $F_v/F_m$ was significantly affected by the light regime, and plants cultivated under R90B10 had the lowest value (Table S5; see also Figure 5). In this species, the non-photochemical quenching (NPQ) and performance index for the photochemical activity ($PI_{ABS}$) parameters were not affected by the treatments (Table S5).

In *Guzmania*, $F_v/F_m$ was not affected by the treatments (Table S6; see also Figure 5). In this species, NPQ was affected by both the light and EC regimes, with the lowest values noted under R80B20 and EC2 (Table S6). In $PI_{ABS}$, an interaction between light and EC regimes was apparent, where plants grown under R90B10 at EC2 having the lowest value (Table S6).

In *Vriesea*, $F_v/F_m$ was significantly affected by the light regime, and plants cultivated under R90B10 had the lowest value (Table S7; see also Figure 5). In this species, NPQ and $PI_{ABS}$ were not affected by the treatments (Table S7).

### 3.6. Photosynthetic Efficiency in Response to Light Intensity

In *Aechmea*, chlorophyll fluorescence obtained at different light intensities (100, 200, 300, 500, and 1000 $\mu$mol m$^{-2}$ s$^{-1}$) was not affected by the treatments (Table S5).

In *Guzmania*, some differences in chlorophyll fluorescence between light quality regimes were apparent, though these were not consistent among measurement light intensities (Table S6). Instead, decreased fluorescence was consistently noted in plants cultivated under higher EC at 100 to 500 $\mu$mol m$^{-2}$ s$^{-1}$ measurement light intensities (Table S6).

In *Vriesea*, some differences in chlorophyll fluorescence among light quality treatments were noted, but these were not consistent among measurement light intensities (Table S7).



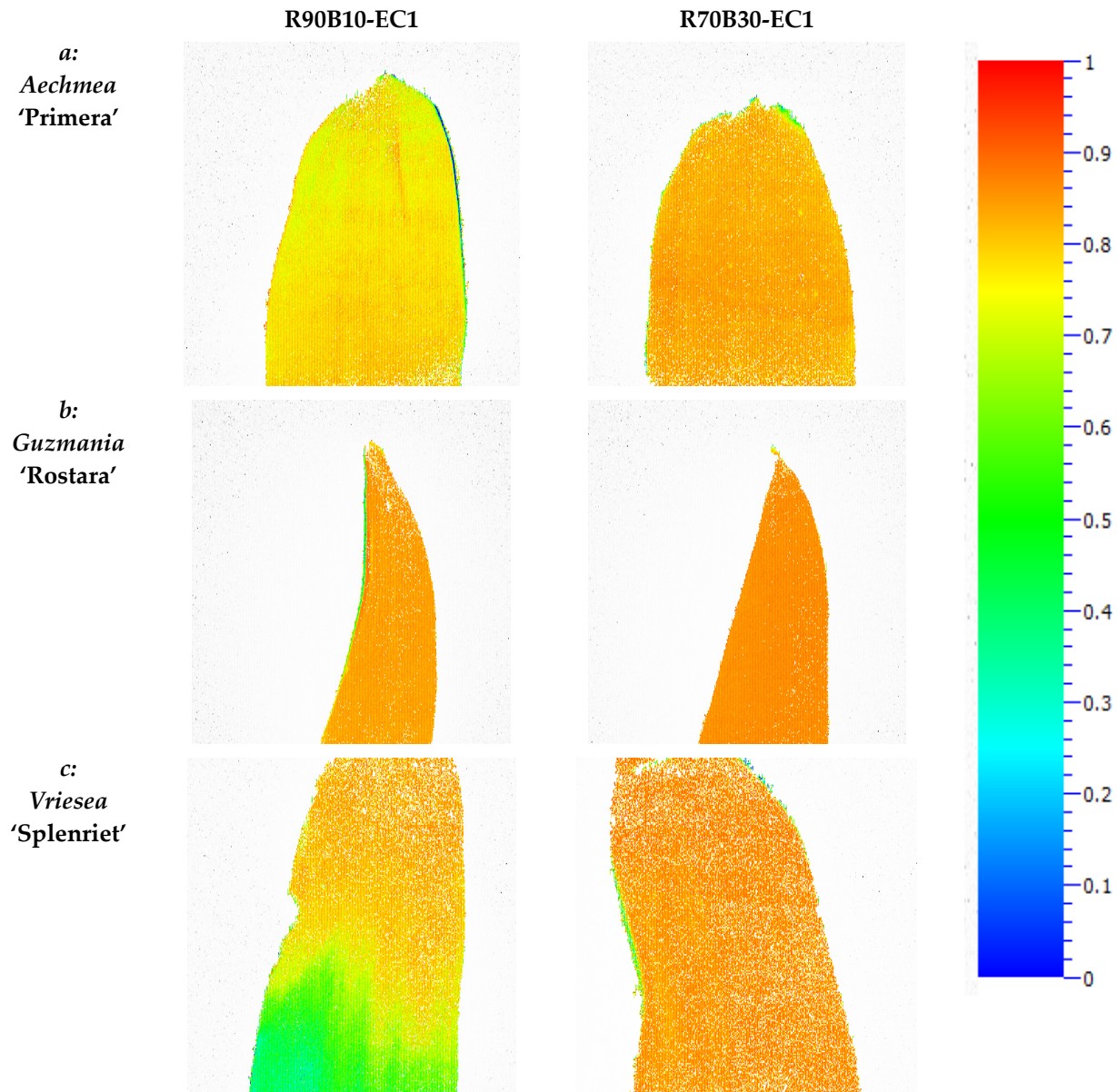

**Figure 5.** Representative images of maximum quantum efficiency of PSII in the three species under study cultivated under supplementary light of R90B10 [90% red (R) and 10% blue (B)] or R70B30 (70% R and 30% B) and nutrient solution electrical conductivity of 1 dS m$^{-1}$ (EC1).

## 4. Discussion

### 4.1. The Optimum EC for Growth Mostly Depends on the Species Rather than on Light Level

In *Aechmea*, the leaf osmotic potential response to the higher EC depended on the light regime (Table S2). In the absence of supplementary light, for instance, *Aechmea* leaf osmotic potential did not differ between the two EC levels. In *Guzmania* and *Vriesea*, by contrast, the higher EC-induced leaf osmotic potential depression was evident independently of the light regime (Tables S3 and S4). These results might be taken to suggest that the overall effect of higher EC on cell turgor pressure was species dependent.

Despite decreased tank volume, higher EC promoted growth (i.e., leaf and flower dry weight; Table 2), and inflorescence development (Figure S2) in *Aechmea*. In this species, increased nitrogen level was earlier associated with enhanced plant growth [36]. Increased urea fertilization also stimulated growth in two other bromeliad species (*Tillandsia pohliana*, *Vriesea philippocoburgii*) by balancing the levels of cytokinins and auxins [37]. On the

contrary, higher EC decreased growth in *Guzmania* (tank volume, root, and flower dry weight; Table 3) and *Vriesea* (leaf area, leaf, root, and flower dry weight; Table 4), and negatively affected inflorescence development (Figure S2). Therefore, the EC level optimum for growth and flowering is clearly species dependent.

Supplementary light slightly alleviated the high EC induced negative effects in *Guzmania* and *Vriesea*. In these species, the higher EC-induced yellow spots on the leaves were only apparent in the absence of supplementary light. Incidence of yellow spots on *Guzmania* leaves owing to elevated EC and low light intensity were also earlier noted [27]. With this minor exception, the present results provide little support that the optimal EC level depends on the light intensity during cultivation of bromeliads.

*4.2. Increased Supplementary Light B Fraction Shortens the Time to Commercial Development Stage*

Supplementary light promoted external quality aspects depending on the species, including plant height, offshoot number, crown thickness, tank volume, and leaf area (Tables 2–4). In all three species, it increased root dry weight, flower dry weight and fresh mass partitioning to the inflorescences (Tables 2–4 and Figure S1). The positive effect of supplementary light on bromeliad plant growth was earlier reported [14,26]. For the first time, this study indicates that both flower dry weight and fresh mass partitioning to the inflorescences were generally enhanced when the proportion of B light in the spectrum was increased (R80B20 and R70B30 as compared to R90B10; Tables 2–4 and Figure S1). Therefore, supplementary light quality is a critical determinant of ornamental value, with enhanced proportion of B leading to superior external quality plants.

Throughout the production–distribution chain (e.g., nurseries, wholesalers), leaf coloration is generally employed as an index indicative of pot plant vigor and health status [30,31]. In all three species under study, the highest proportion of B light in the spectrum (R70B30) was generally associated with increased SPAD value (Tables S2–S4). In *Aechmea*, leaf chlorophyll was also significantly increased under R70B30 (Table S2). Therefore, supplementary light with the highest proportion of B light in the spectrum (R70B30) additionally promotes pot plant ornamental value by increasing leaf coloration.

Under natural light, the period required for inflorescence emergence was 47.5, 55.9, and 76.3 d for *Aechmea, Guzmania* and *Vriesea*, respectively (Figure 3). Supplementary light decreased this period (Figure 3). Earlier studies also indicated accelerated flowering owing to enhanced light intensity [14,26]. We additionally show here that when an increased proportion of B light in the spectrum (R80B20 and R70B30 as compared to R90B10) was employed, the time required for inflorescence emergence was further reduced (Figure 3).

Supplementary light also increased inflorescence development (Figure 4). This is in accordance with earlier findings [14,26]. Notably, an increased proportion of B light in the spectrum (R80B20 and R70B30 as compared to R90B10) promoted inflorescence development in *Aechmea* and *Guzmania* (Figure 4). In this way, the time to commercial development stage can be considerably shortened.

Under supplementary light, photosynthesis rate is higher [38]. This enhanced rate is associated with increased light level, though improved photosynthetic efficiency may be a contributing factor [10,11]. In this study, the functionality of photosynthetic apparatus was assessed by both $F_v/F_m$ imaging (providing spatial pattern) and OJIP test. Results obtained by either protocol indicated that supplementary light generally induced minor effects on photosynthetic apparatus state (Tables S5–S7). A notable exception to this trend was the negative effect of R90B10 on $F_v/F_m$ in *Aechmea* and *Vriesea* (Tables S5 and S7; see also Figure 5). Therefore, the noted differences in growth and flowering were not related to variation in photosynthetic efficiency.

## 5. Conclusions

A greenhouse study was conducted to elucidate the effect of supplementary light quality and nutrient solution EC on plant growth, and inflorescence development in three bromeliad species. Light regimes included solar light, and this supplemented with R90B10

(90% R and 10% B), R80B20 (80% R and 20% B), and R70B30 (70% R and 30% B). EC was set to 1 or 2 dS m$^{-1}$. The higher EC promoted growth, as well as inflorescence emergence and development in *Aechmea*. By contrast, it induced negative effects in *Guzmania* and *Vriesea*. In these two species, supplementary light slightly alleviated the higher EC-induced negative effects. With few notable exceptions, supplementary light generally induced minor effects on the photosynthetic apparatus state. Depending on the species, supplementary light improved decorative features (plant height, offshoot number, crown thickness, tank volume, and leaf area). In all three species, supplementary light improved root and inflorescence weight, and promoted biomass allocation to generative organs. Importantly, it also sped up inflorescence emergence and improved inflorescence development. In this perspective, the duration needed for commercial ripeness was considerably reduced. These effects were more prominent under an increased proportion of B light in the spectrum (R80B20 and R70B30 as compared to R90B10). Supplementary light with increased B fraction results in a shorter production cycle owing to the more rapid emergence and enhanced development of the inflorescence and is highly recommended for commercial use.

**Supplementary Materials:** The following are available online at https://www.mdpi.com/article/10.3390/horticulturae7110485/s1, Figure S1. Time to inflorescence emergence following acetylene treatment as a function of nutrient solution electrical conductivity (EC) in the three species under study. Data of the four light treatments were pooled. A total of 16 replicate plants were assessed per treatment. Error bars represent SEM, Figure S2. Inflorescence development as a function of nutrient solution electrical conductivity (EC) in the three species under study. Data of the four light treatments were pooled. The scale (1 to 5) characterizing inflorescence development is provided in Figure 1. A total of 16 replicate plants were assessed per treatment. Error bars represent SEM, Table. S1: effect of supplementary light and nutrient solution electrical conductivity (EC) on dry mass partitioning of the three species under study, Table S2: effect of supplementary light and nutrient solution electrical conductivity (EC) on leaf osmotic potential, photosynthetic pigment content and SPAD value of *Aechmea* 'Primera' plants, Table S3: effect of supplementary light and nutrient solution electrical conductivity (EC) on leaf osmotic potential, photosynthetic pigment content and SPAD value of *Guzmania* 'Rostara' plants, Table S4: effect of supplementary light and nutrient solution electrical conductivity (EC) on leaf osmotic potential, photosynthetic pigment content and SPAD value of *Vriesea* 'Splenriet' plants, Table S5: effect of supplementary light and nutrient solution electrical conductivity (EC) on leaf photosynthetic functioning of *Aechmea* 'Primera' plants, Table S6: effect of supplementary light and nutrient solution electrical conductivity (EC) on leaf photosynthetic functioning of *Guzmania* 'Rostara' plants, Table S7: effect of supplementary light and nutrient solution electrical conductivity (EC) on leaf photosynthetic functioning of *Vriesea* 'Splenriet' plants.

**Author Contributions:** E.J.A., S.A., N.A. and M.R.R. conceived and designed research. E.J.A. and M.S. conducted experiments, handled material preparation, and performed data collection. E.J.A., S.A., N.A., M.R.R. and D.F. analyzed data and make interpretation. D.F., E.J.W. and G.T. com-mented on data analysis. E.J.A., S.A., M.S. and D.F. wrote the manuscript. D.F., E.J.W. and G.T. revised the manuscript. All authors have read and agreed to the published version of the manuscript.

**Funding:** This research received no external funding.

**Institutional Review Board Statement:** Not applicable.

**Informed Consent Statement:** Not applicable.

**Data Availability Statement:** Raw data are available upon request from the corresponding author.

**Acknowledgments:** We are grateful to Khadem nursery for the donation of the plant material, and to Mahboobeh Zare Mehrjerdi, Fardad Didaran, Mohsen Ghadermarzi, and Mahsa Karimzadeh for their invaluable contribution in designing and conducting the experiment. The valuable comments of the editor and three anonymous reviewers are greatly appreciated.

**Conflicts of Interest:** The authors declare no conflict of interest.

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
