# Peer review of "Supplementary Light with Increased Blue Fraction Accelerates Emergence and Improves Development of the Inflorescence in Aechmea, Guzmania and Vriesea"

_horticulturae, doi:10.3390/horticulturae7110485_

Round 1

Reviewer 1 Report

Supplementary light with increased blue fraction shortens the production period in Aechmea, Guzmania and Vriesea due to accelerated emergence and improved development of the inflorescence

This report contributes to a better understanding of the flowering biology and plant development of three bromeliads that are important in ornamental horticulture. The research shows that both nutrition and light quality contribute to the regulation of the developmental processes and can be used in the improved production protocols of these species.

I suggest revision of this paper with my comments presented in the attached file. The main points are as follows:

  1. This paper looks like an MSc thesis, in which a student presents all the data, even if not significant or important. I suggest revising the text and omitting all non-conclusive graphs.
  2. The Introduction does not relate to several important aspects:
    1. What are the "tank species"?
    2. What does the Ref #8 (in Dutch) relate to? You compare your data with this Ref in the Discussion, but this Ref is not in English and so some readers are unable to read this paper and thus understand its significance.
    3. Why do you need to mention the metabolic types of the species on page 2 if you never refer to them again?
    4. I suggest adding information on flower induction in the Introduction
  3. Non-conclusive graphs , e.g., Fig. 6, can be omitted and the information presented in the text
  4. The Discussion relates mainly to a comparison between Reference #8 and #17. I think that larger biological discussion and comparison with other species will contribute to the scientific meaning of the paper.
  5. Please avoid repetitions between the Discussion and Conclusions. I am not sure that you need the Conclusions at all.

Author Response

Reviewer #1

This report contributes to a better understanding of the flowering biology and plant development of three bromeliads that are important in ornamental horticulture. The research shows that both nutrition and light quality contribute to the regulation of the developmental processes and can be used in the improved production protocols of these species.

I suggest revision of this paper with my comments presented in the attached file. The main points are as follows:

(1) This paper looks like an MSc thesis, in which a student presents all the data, even if not significant or important. I suggest revising the text and omitting all non-conclusive graphs.

Authors: As suggested, Figures 3 and 6 were moved to Supplementary Material (new Figures S1 and S2). In this revised version, the manuscript figures include only the key findings.

(2) The Introduction does not relate to several important aspects:

(2.1) What are the "tank species"?

Authors: This is now clarified (Lines 94–96).

(2.2) What does the Ref #8 (in Dutch) relate to? You compare your data with this Ref in the Discussion, but this Ref is not in English and so some readers are unable to read this paper and thus understand its significance.

Authors: In the species under study, this Dutch paper is the only one. There is no English literature available.

(2.3) Why do you need to mention the metabolic types of the species on page 2 if you never refer to them again?

Authors: The motivation of using species employing different types of photosynthesis is now provided (Lines 91–96). The noted differences in growth were not related to photosynthetic efficiency (Lines 426–427).

(2.4) I suggest adding information on flower induction in the Introduction

Authors: The use of ethylene releasing agents to control flowering in commercial settings was now added (Lines 59–60).

(3) Non-conclusive graphs, e.g., Fig. 6, can be omitted and the information presented in the text.

Authors: As suggested, Figures 3 and 6 were moved to Supplementary Material (new Figures S1 and S2). In this revised version, the manuscript figures include only the key findings.  

(4) The Discussion relates mainly to a comparison between Reference #8 and #17. I think that larger biological discussion and comparison with other species will contribute to the scientific meaning of the paper.

Authors: No literature related to the effect of light quality on the flowering of non-photoperiod species is in existence. The present study is the first one.

(5) Please avoid repetitions between the Discussion and Conclusions. I am not sure that you need the Conclusions at all.

Authors: Several sentences were re-written, and the conclusion section was reduced.

Comments within the pdf file

(6) Line 27: Tank bromeliads have leaves that form a reservoir to hold water at their bases- but it is not a well-known term. I suggest to move it in the introduction, and explain.

Authors: As suggested, this term was erased from the Abstract, and moved to the Introduction, where it is explained (Lines 94–96).

(7) Line 46: on bromeliads

Authors: Done (Line 44).

(8) Line 47: erase

Authors: As suggested, this sentence was erased.

(9) Line 50: in all species

Authors: Done (Line 48).

(10) Lines 95–96: please relate to these differences in the introduction. Why you mention them? How this affects your studies? References

Authors: (I) The motivation of using species employing different types of photosynthesis is now provided (Lines 91–93). (II) For each species, the respective references were also added as related to the type of photosynthesis (Lines 99–101). (III) The noted differences in growth were not related to photosynthetic efficiency (Lines 426–427).

(11) Lines 99–102: English editing

Authors: Language was revised (Lines 103-104).

(12) Line 107: range is large, omit average, I suppose that this is not fully controlled greenhouse

Authors: Done (Lines 111–112).

(13) Lines 108–110: move to discussion or omit

Authors: As suggested, these lines were erased.

(14) Line 118–119: 0800 to 2000. Did it change photoperiod?

Authors: This is now provided (Lines 120–122).

(15) Lines 141–143: I strongly suggest to relate flower induction by ethylene in the introduction. Otherwise, it looks too technical.

Authors: The use of ethylene releasing agents to control flowering in commercial settings was now added (Lines 59–60).

(16) Line 194: do you mean SPAD that measures leaf transmittance in red light at 650 nm (at which chlorophyll absorbs) and in near-infrared light at 940 nm (for the correction of leaf thickness). The ratio of these two transmission values is referred to as SPAD reading or SPAD value (Hoel and Solhaug 1998). Please clarify.

Authors: This is now clarified, and the suggested reference was added (Lines 200–202).

(17) Lines 293–294: I suggest to omit this fig or move it to suppl. information can be delivered in the text

Authors: As suggested, Figure 3 was moved to Supplementary Material (new Figure S1). In this revised version, the manuscript figures include only the key findings.

(18) Lines 297–298: see my comment in Figure 6, not sure if this statement is correct.

Authors: As suggested, Figure 6 was moved to Supplementary Material (new Figure S2). In this revised version, the manuscript figures include only the key findings. The statistics supporting this sentence are provided in Table S1.

(19) Lines 320–322: sorry, all looks non-significant to me. I feel that this figure can be deleted and the results reported in the text.

Authors: As suggested, Figure 6 was moved to Supplementary Material (new Figure S2). In this revised version, the manuscript figures include only the key findings. The respective statistics are provided in Table S1.

Reviewer 2 Report

Dear Editor, Dear Authors,

thank you for considering me as a reviewer of the manuscript entitled: "Supplementary light with increased blue fraction shortens the production period in Aechmea, Guzmania and Vriesea due to accelerated emergence and improved development of the inflorescence".

I believe that the subject of research presented in this manuscript is rare, but very interesting and of great practical importance in the production process of Aechmea, Guzmania and Vriesea. All methods that shorten the period of cultivation of ornamental plants contribute to lowering production costs.

The research was carried out correctly and the results are presented in an interesting way. However, I have some suggestions for improving this manuscript:

1) The manuscript title is too long, it should be edited and shortened.

2) Abstract is correctly written and contains all the necessary elements, but is too detailed and too long.

3) The Introduction chapter is poorly developed. The authors cite only 14 publications here. I believe that the introduction to the subject of research should be developed.

4) References - in general, the authors cite only 24 publications in the manuscript, which is definitely not enough. Please expand the scope of the literature cited.

5) The entire manuscript should be re-edited in order to bring it into line with the current TEMPLATE. Please pay particular attention to the References section.

Overall,  in my opinion, the manuscript presents valuable results that merit publication in the Horticulture journal.

Author Response

Reviewer #2

Dear Editor, Dear Authors,

thank you for considering me as a reviewer of the manuscript entitled: "Supplementary light with increased blue fraction shortens the production period in Aechmea, Guzmania and Vriesea due to accelerated emergence and improved development of the inflorescence".

I believe that the subject of research presented in this manuscript is rare, but very interesting and of great practical importance in the production process of Aechmea, Guzmania and Vriesea. All methods that shorten the period of cultivation of ornamental plants contribute to lowering production costs.

The research was carried out correctly and the results are presented in an interesting way. However, I have some suggestions for improving this manuscript:

(1) The manuscript title is too long, it should be edited and shortened.

Authors: As suggested, the tile was shortened (Lines 2–4).

(2) Abstract is correctly written and contains all the necessary elements, but is too detailed and too long.

Authors: As suggested, the abstract was shortened by removing several details.

(3) The Introduction chapter is poorly developed. The authors cite only 14 publications here. I believe that the introduction to the subject of research should be developed.

Authors: As suggested, the Introduction was enriched, and several references were now included (Lines 59–61, 91- 96).

(4) References - in general, the authors cite only 24 publications in the manuscript, which is definitely not enough. Please expand the scope of the literature cited.

Authors: As suggested, several relevant references were now added (Lines 59–61, 91-96, 100–101, 186, 197, 203, 399,419).

(5) The entire manuscript should be re-edited in order to bring it into line with the current TEMPLATE. Please pay particular attention to the References section.

Authors: As suggested, the manuscript and reference layout was now adjusted to the journal one.

Overall, in my opinion, the manuscript presents valuable results that merit publication in the Horticulture journal.

Reviewer 3 Report

Authors gave results which are practical and will be useful in the horticulture of bromeliads. Ms is not scientifically perfect but from gardening point of view should be publish in Horticulturae after some changes:

Please give full Latin names of studied species to abstract. All genera names in Latin must be written using italics (correct in whole text).

Please provide, data about lighting and flowering period of studied species from natural habitat (use data from literature) and compare with your results.

Author Response

Reviewer #3

Authors gave results which are practical and will be useful in the horticulture of bromeliads. Ms is not scientifically perfect but from gardening point of view should be publish in Horticulturae after some changes:

(1) Please give full Latin names of studied species to abstract. All genera names in Latin must be written using italics (correct in whole text).

Authors: This was corrected throughout the manuscript.

(2) Please provide, data about lighting and flowering period of studied species from natural habitat (use data from literature) and compare with your results.

Authors: As suggested, data on the period required for flowering under natural habitat conditions compared with our results.  (Line: 404-410)
